# Knowing and the Known: A Philosophical and Pedagogical Critique on the Concept of 'Powerful Knowledge'

Siebren Miedema [1,2] 

1   Faculty of Behavioural and Movement Sciences, Vrije Universiteit Amsterdam,
    1081 BT Amsterdam, The Netherlands; s.miedema@vu.nl
2   Faculty of Religion and Theology, Vrije Universiteit Amsterdam, 1081 BT Amsterdam, The Netherlands

**Abstract:** It is remarkable how the popularity of the concept of 'powerful knowledge' has increased during the last decade in academic circles and among politicians, too. This is especially the case when the issue of the place and function of knowledge in the curriculum is addressed. A strong impetus for the increased attention paid to this concept of knowledge came from the writings of Michael Young and Johan Muller. Based on his own critical-hermeneutical-pragmatist-and-(neo-)Vygotskian-inspired philosophy of education and philosophy of science as his 'Vorverständnis' (Gadamer), but also based on the recent criticism articulated by the philosophers of education John White and Ingrid Carlgren and educational theorist Gert Biesta, the author shows the philosophical, pedagogical and didactical inadequacy of this concept. The author is criticizing the philosophical and pedagogical presuppositions of Young and Muller's stance in propagating their core concept of 'powerful knowledge' as it is grounded in a social realist view and the way this concept has been used in educational studies by others. It is the author's conviction that the concept of 'powerful knowledge' and the underlying social realist paradigm are incompatible and even incommensurable (in a Kuhnian sense of the terms) with sociocultural and pragmatist paradigms. It is, in his view, theoretically and conceptually confusing when authors who work along the lines of these paradigms are trying to complement these with the concept of 'powerful knowledge' along the lines of social realism as outlined by Young and Muller. Let us stick to knowing and the known as a theoretical conceptualization of 'knowledge'.

**Keywords:** powerful knowledge; knowing and the known; social realist view; pragmatism; social–cultural theory; incommensurability

## 1. Introduction

It is really striking how the popularity of the concept of 'powerful knowledge' has increased during the last decade among academics as well as politicians. This is especially the case when the issue of the place and function of knowledge in the curriculum is addressed. A strong impetus for this popularity came from the writings of Michael Young (e.g., Young 2013) and Johan Muller (e.g., Muller and Young 2019). Young and Muller rightly state that discussions about skills and competencies have been prioritized to the detriment of debates on the positioning of knowledge in the curriculum. They have brought the concept of 'knowledge' back into debates with their favorite core concept, 'powerful knowledge'.

The crucial question, in my view, is, however, whether their approach can stand criticism from the perspective of the philosophy of education, educational theory, curriculum theory and the philosophy of science. The concept of 'powerful knowledge', as it is grounded in a social realist view, has been severely criticized by the philosophers of education John White (2018, 2019) and Ingrid Carlgren (2020). The educational theorist Gert Biesta has criticized Young's mainly Durkheimian-inspired view that pragmatism is a

relativistic form of antirealism. He favors John Dewey's theory of knowing, his transactional realism, contrary to Young's social realism. It is Biesta's conviction that knowledge should indeed be brought back into the curriculum, however, via Deweyan pragmatism, and he makes clear what the theoretical and practical implications are for the use of the concept of 'knowledge'.

Curious enough, there are others who have tried to bridge, for example, the different positions of Young and White (Reiss 2017), the positions of Young and Biesta (Nordgren 2017) or the positions of Young's social realism with social–cultural theories in the tradition of Vygotsky and Bakhtin and Deweyan pragmatism (Osbeck et al. 2023; see also Sporre 2023).

It is theoretically and practically not an easy task to make sense of these different trains of thought when we pose the crucial question: "How to bring knowledge back in the curriculum and in a way that is philosophically, pedagogically and didactically consistent, compatible and commensurable?".

In the remainder of this contribution, I present in Section 2 the method used in this essay. I will then present the separate results of my analysis in Sections 3–5, respectively, in relationship with the view of Young, and Muller and Young, on powerful knowledge (Section 3), with the critical stances related to Young and Muller's and Young's use of powerful knowledge (Section 4), and in Section 5 in respect to some mediating attempts of conflicting positions in the debates on powerful knowledge. In Section 6, the results of my reconstructive exercise are presented. Finally, I will end in Section 7 with a few short points for discussion.

## 2. The Method Used and the Author's 'Vorverständnis' of Some Theoretical Core Concepts

The focus in this contribution is, as might already be very clear from the title, theoretical and pedagogical. From the perspective of the philosophy of education and the philosophy of science, I use the method of conceptual or reconstructive analysis to make insightful which connotations of concepts and/or conceptions are used (Wilson 1963; Steutel 1991; Van der Kooij 2016). Leading in this was the conceptual use of the notion 'powerful knowledge' and the formulated criticism on using this term in different and mostly theoretical articles. In empirical terminology, selectively chosen representative articles or chapters analyzed form the data for my analysis, and the results are the qualitative presentations of my conceptual and theoretical analysis in Sections 3–5. Criteria that were used in the analysis were conceptual clarity and conceptual consistency. Along these lines, I can do justice to argumentative reliability as part of the overarching fundamental methodological criterion of intersubjective comprehensibility (Van IJzendoorn and Miedema 1986).

From the outset, I want to make clear how I myself conceptualize the interrelatedness of the notions of 'pedagogy', 'curriculum' and 'didactics' that are presenting my theoretical 'Vorverständnis' (Gadamer 1960). With the concept of 'pedagogy', I mean Bildung (edification/formation) in an encompassing sense; that is, in terms of Wolfgang Klafki, at the individual level, aiming at self-reliance, self-responsibility and self- and codetermination, and socially, Bildung is directed at sociability and solidarity (Klafki 1970, p. 26; Miedema and Bertram-Troost 2014). Pedagogy forms with this core concept of Bildung the horizon for the intertwined concepts of curriculum and didactics. Thus, there is an intrinsic conceptual and practical relationship between pedagogy, curriculum and didactics.

## 3. The Young–Muller View on Powerful Knowledge

In his 2013 article, Young presents two characteristics of powerful knowledge applicable both to STEM (science, technology, engineering and mathematics) as well as to SSH (social sciences and humanities). Firstly, it is specialized knowledge in how it is produced in academia by specialized disciplines. It is thus a discipline-based, knowledge-forming system of interrelated concepts. It is also specialized in how it is transmitted in educational settings (schools, colleges and universities). Secondly, powerful knowledge differs from the experiences that pupils and older learners bring to their respective educational

settings. There are hard conceptual boundaries between school and everyday knowledge (Young 2013, p. 108). Powerful knowledge as being conceptual knowledge is recontextualized knowledge. "Re-contextualization . . . means . . . the selection, sequencing and pacing of contents that takes into account both the coherence of the discipline subject and the limits on what can be learned by students at different stages of their development" (Young 2013, p. 109). Young is contrasting recontextualized knowledge with everyday knowledge that is based on context-specific experience. Such latter kind of knowledge will enable the children to make sense of the world and is always related to very specific contexts. "(T)he curriculum refers to knowledge that pupils are entitled to know; it does not include pupils' (everyday, SM) experiences (. . .) (P)upils do not come to school to know what they already know from experience" (Young 2013, p. 111). So, the focus of the curriculum, according to Young, is on the pupils and their acquisition of knowledge, and the focus of pedagogy is on what teachers do. "(It) depends on both the knowledge that teachers have of their subject, the knowledge that they have about individual pupils and how they learn—and the knowledge that informs what they require their pupils to do" (Young 2013, p. 111).

In their 2019 article, Muller and Young have, as they coined it, revisited their core characterization of powerful knowledge as forming systems of interrelated academic concepts. It is their intention to retain "a satisfactory account of 'powerful knowledge'; and . . . to provide a satisfactory account of the 'power' of knowledge in the humanities" (Muller and Young 2019, p. 196). This is, in their view, necessary because powerful knowledge has turned out to be "a rather multifaceted educational object" (Muller and Young 2019, p. 207). That is why they are now treating the powerfulness of knowledge in a different way "as 'potential or capacity' for social actors to do something (. . .) (W)e understand this not only in the restricted sense of a capacity to act, but also in the socio-epistemic senses in which it embodies positive or transformative power or potential power" (Muller and Young 2019, pp. 208–9). For this they use Spinoza's distinction between potestas as power over, and potentia as power to, that is, the ability and capacity to do something. "(P)otentia is productive or creative, it extends horizons, it images new futures . . . involves the capacity to achieve something of value. In this sense highly specialized knowledge as produced by universities confers a very specialized capacity to its holders" (Muller and Young 2019, pp. 201–2).

Muller and Young distinguish three meanings of 'powerful knowledge'. They first emphasize again its character as disciplinary-related power with respect to "disciplines as a community of self-governing peers . . . (that) produce specialized discourses that regulate and ensure reliability, revisability, and emergence" (Muller and Young 2019, p. 209) and to "(d)isciplinary meaning . . . that is generative, in that it establishes an indirect relation of meaning between the concept and an aspect of the world" (Muller and Young 2019, p. 209). Secondly, there is the relationship between the curriculum content and the respective disciplines. There is, thirdly, the generative power at the side of the pupils; that is, them being able "to make new connections, gain new insights, generate new ideas" (Muller and Young 2019, p. 210).

It seems that using the notions of 'socio-epistemic' and 'transformative' in this revitalized approach can offer openings for other than their own social realist paradigmatical position.

## 4. The Critics

John White has criticized in two articles the notion of powerful knowledge. White is a representative of the British philosophy of education tradition and well versed in the analytical tradition. In the 2018 article, he is bringing to the fore the practical turn in the analytical tradition, more specifically in the work of Paul Hirst around the 90s. He criticizes the notion of powerful knowledge characterized as systems of interrelated concepts and the equation of the concept of 'powerful knowledge' with the Chartrists' notion of 'really useful knowledge'.

He further argues that Young's approach has the same shortcomings as Hirst's theoretical notion of 'forms of knowledge' in Hirst's writings before the 90s. They both make "systems of *sui generis*, interrelated concepts the criterion for selecting central curriculum areas. Another is the link made with the defining significance, in the case of human beings, of the use of (symbolic) concepts" (White 2018, p. 332). Hirst's basic idea was that "there are seven or eight distinct, and together comprehensive, forms of human understanding. Inducing students into them, not for extrinsic reasons but as an end in itself, is developing the rational mind that we possess as a uniquely human attribute" (White 2018, p. 331). Interestingly, White points to the fact that Young has criticized Hirst's forms of knowledge approach in 1971 himself.

A crucial difference, however, between Young and (the early) Hirst is that for Young, there is a sharp distinction between everyday knowledge and disciplinary conceptual knowledge, while Hirst saw disciplinary forms of knowledge as embryonically embedded in everyday knowledge (White 2018, p. 332).

In his 2019 article, White has made it clear that he is not convinced that the new approach along the line of potential can rescue the use of the notion 'powerful knowledge'. To his embarrassment, there has been a complete conceptual shift because they have fundamentally altered the descriptive meaning of the term. However, and most importantly for White, powerful knowledge is also in this revisited approach "still knowledge pursued and taught by specialized disciplinary academic groups such as mathematicians, biologists, geographers, historians and experts in literature" (White 2019, p. 432). It is still prioritizing academic theory over practice, over action and over experience, as embodied in the lifeworld. So, his plea is to completely abandon the use of the phrase 'powerful knowledge' and to use "terminology appropriate to impartial scholarly investigation rather than language more at home in the world of product promotion" (White 2019, p. 437) and just stick to the term 'specialized knowledge' because it "carries no in-built assumption that it is a good thing" (White 2019, p. 436).

In her impressive article, Ingrid Carlgren is presenting a strong philosophical philosophy of education, as well as a didactically underpinned pragmatist alternative in criticizing the cartesian, rationalist and dualistic view of Muller and Young, who give priority to theory over practice (Carlgren 2020). She states that they probably have completely missed the practice turn in social theory as well as the practice turn in learning theory in the 1980s and 1990s. Carlgren is combining a sociocultural (neo-Vygotskian) perspective with a Deweyan perspective (see also Carlgren et al. 2015). The latter perspective is especially inspired by the 1949 Dewey–Bentley transactional view on knowing and the known (Dewey and Bentley 1949). Carlgren is following this Dewey–Bentley transactional train of thought, and for her, 'powerful knowing' is the core notion. "The concept of knowing connects the known to the world of human activity" (Carlgren 2020, p. 225). "In contrast to the known which is explicit and can stand by itself independent of anyone knowing it (compare this with Popper's world 3, Popper 1972, pp. 106–52; that is, knowledge without a knowing subject, without a knower, SM), the knowing refers to someone who knows. Knowing is connecting the known to a knower as well as to the context where the known is functional in some way. The knowing thus points in two directions: on the one hand to the known, and on the other to the person as well as the activity where it is used" (Carlgren 2020, p. 331).

"If the aim of education is to produce knowledgeable people (on page 331, she is using the pedagogical notion of 'development' instead of 'produce' and states that "(t)he object of teaching is to develop the pupils' knowing of specific knowns", NN), it is the knowing of the known that is primary in teaching (. . .). The differentiation between the knowing and the known opens to explore the knowings within different specialized knowledge areas" (Carlgren 2020, p. 332). She relates this also to the conceptualization of Michael Polanyi's concept of 'personal knowledge', in which he is using the concepts of knowledge and knowing interchangeably, however, with a preference for the concept of knowing because this emphasizes much better the personal aspect of knowledge (Polanyi 1962). And according to Carlgren, this personal aspect of knowing that means that knowledge

is borne by people is also prominently present in schools and in the work of teachers, as well as central in Bildung-related theories, combining "the idea of knowledge and personal development in a way that corresponds to the two-sidedness of the concept of Bildung" (Carlgren 2020, p. 232).

With this conceptualization of knowing and the known, Carlgren wants to get rid of the unfruitful dichotomies of 'knowing that' and 'knowing how' or of theoretical knowledge and practical knowledge. Although she does not go so as far as to completely abandon the use of the notion 'powerful knowledge' (Carlgren 2020, p. 324), her criticism of Muller and Young's stance and her own preference for using the pragmatist interrelatedness of knowing and the known points is, in my view, clearly in a rejecting direction.

The educational theorist Gert Biesta has criticized Young's mainly Durkheimian-inspired view on pragmatism as a relativistic form of antirealism (Biesta 2014). He is presenting, in a thorough reconstruction, John Dewey's theory of knowing, his transactional realism, contrary to Young's social realism, as a third way next to objectivism and relativism. In Dewey's view, to acquire knowledge, action is a necessary prerequisite. However, it is not a sufficient one because only the combination of action and reflection will result in knowledge (see on Dewey's position also Miedema and Biesta 1994).

But Biesta is doing much more in terms of pedagogy and curriculum. He outlines in extenso the consequences of a Deweyan train of thought with the core notions of 'experience', 'reality', 'knowledge' and, pedagogically and didactically also very relevant, the notion of 'coordination'. From a Deweyan pedagogical view, 'coordination' has to do with the adequate relationship between individual and social–societal factors, the relationship between child and culture (Dewey 1938). Dewey is criticizing a 100% child-centered approach as well as a 100% content- or tradition-centered approach. The carefully pedagogically chosen subject matter, the selection out of the traditional and culturally loaded stock of content, is not meant to be just transmitted and simply reproduced by the pupils, but it is the developmental stuff that has potential for Bildung, it has transformative power for them (Miedema and Bertram-Troost 2014). In this very process also, (the meaning) of the tradition is renewed (Dewey 1916, p. 1).

Dealing with this notion of 'coordination', Biesta is also criticizing Young's Dewey interpretation that Dewey was part of a progressive, learner-centered tradition (Young 2013, p. 102). In his classic and most concise publication of his philosophy of education from 1938, *Experience & Education* (Dewey 1938), Dewey is precisely criticizing the one-sidedness of progressive education as well as of traditional education because neither of the two applies the principles of a carefully developed philosophy of experience. It is Dewey's contention, as we have already seen, that the core problem of pedagogy is the coordination of the child and the curriculum, in other words, the coordination of individual (personal) and societal (cultural) factors. That is why 'experience' functions both as a mean and as an aim. Biesta's conclusion is, however, very clear regarding the use of the notion of 'powerful knowledge': knowledge should indeed be brought back into the curriculum, however, via Deweyan pragmatism.

## 5. The Mediators

In his contribution to the curriculum discussion in general and the one on powerful knowledge in particular, *Michael Reiss* has tried to bridge the different positions of Young and White (Reiss 2017). It is his conviction that instead of applying Young's ideas inflexibly or naively but sensitively, they have the potential to complement the work of White (Reiss 2017, p. 130). With the severe criticism of White on Young's view, with his 2018 and 2019 articles in mind, as presented above, especially his plea to completely abandon the use of the phrase 'powerful knowledge', it is interesting to learn the underpinning of his optimism.

Reiss points to Young's use of Vygotsky's distinction between everyday concepts and scientific concepts and Young's own distinction between everyday knowledge and experience and powerful, that is, academic knowledge. A core question in his comparison

is, what place does everyday knowledge have in powerful knowledge? Reiss is referring to Vygotsky's influence on the view of Young. However, unfortunately, Young's Vygotsky interpretation does not hold as is clearly outlined by the Vygotsky experts Van der Veer and Valsiner. Scientific concepts/knowledge and everyday concepts/knowledge and experience are not incompatible but are partners in an intricate interrelationship; they have their foundation in everyday concepts (Van der Veer and Valsiner 1991, pp. 268–75). Vygotsky has also criticized the idea that everyday concepts have been spontaneously invented by the child and has explicitly acknowledged the role of adults in the formation of these so-called spontaneous concepts. That is why Vygotsky preferred to use the notion 'everyday concepts'. This is clearly in line with White's position. Thus, I see no common ground for interpreting the positions of Young and White as complementary, as Reiss has tried to do.

*Kenneth Nordgren* (Nordgren 2017) has tried to reconceptualize the debate on powerful knowledge by relating it to curriculum theory and the three educational purposes Gert Biesta has distinguished, that is, qualification, socialization and subjectification (see, e.g., Biesta 2020). Where Young is very clear on the hard conceptual boundaries between school and everyday knowledge, as we have seen in his interpretation of that relationship, Nordgren is using this distinction in a much softer way and is throwing the angle out of Young's sharp distinction. Also, referring to Biesta's plea for pragmatizing the curriculum and bringing knowledge back into the curriculum conversation but via pragmatism, Nordgren surprisingly concludes that "Biesta's reasoning does not, as I see it, jeopardize the idea of powerful knowledge" (Nordgren 2017, p. 669). It is a pity that Nordgren did not take into account Biesta's extensively articulated pragmatist position along the lines of Dewey in his 2014 article. That article of Biesta, by the way, is also listed in Nordgren's list of references. Seriously dealing with the pragmatist position Biesta has presented in that article might at least have tempered his statement that Biesta is not jeopardizing the use of the notion of 'powerful knowledge' in the sense of Young and Muller and Young.

The chapter by *Osbeck, Sporre and Lilja* is a highly interesting one because the authors bring together Young and Muller's social realist view on powerful knowledge and combine this with social–cultural theories in the tradition of Vygotsky and Bakhtin and Deweyan pragmatism (Osbeck et al. 2023). After briefly introducing the notion of powerful knowledge, they also very briefly deal with the criticism formulated by White (2018) and Carlgren (2020). They accept their criticism, however, without explicitly getting rid of the notion of 'powerful knowledge' further on in the presentation of their theoretical and empirical work.

In the chapter, they present, next to the notion of powerful knowledge, a strong sociocultural approach, especially along the lines of Mark Tappan's work as inspired by Vygotsky and Bakhtin, and combine this with a Deweyan approach using Sven Hartman's inspiring writings and in reference to Biesta (2022). The question is why they still are of the opinion that the concept of 'powerful knowledge' is theoretically compatible with sociocultural and pragmatist approaches. Or shouldn't we just conclude that Young and Muller's social realist paradigm is incompatible, and in Kuhnian terms even incommensurable (Kuhn [1962] 1970), with sociocultural and pragmatist approaches?

In the authors' view, the child and the curriculum are mutually dependent on each other, and they characterize knowledge that draws on that interrelationship as 'powerful knowledge' (Osbeck et al. 2023, p. 22). It may be crystal clear that this conceptualization is completely different from the one used by Young and Muller because here, the experiences of the children are taken seriously next to the input in the curriculum of education, which is culturally loaded content by the teachers, and in their empirical projects, also together with the researchers. And why should this kind of knowledge be characterized as 'powerful'? To elaborate on this, and in line with Biesta's pragmatizing view (Biesta 2014), in Dewey's terminology, the educational process can be defined as the continuing reflexive reconstruction of the experience of the transactional relationship of the individual (or group) and the cultural (and natural) environment. 'Knowledge', then, is the possible relationship between

actions and the consequences of these actions. Insights based on earlier experiences, that is, generalizations, can also be used here next to different views, hypotheses, insights and ideas at hand. The criterion is their (potential) meaningfulness in the practical learning processes and their possible function and role in the reception and interpretation of new experiences gained in different contexts (Dewey [1920] 1948, pp. 149–52). In my view, the authors are adequately able to present and articulate their social–cultural and pragmatist views on their empirical projects and results without any need to use the notion of 'powerful knowledge'. Using this notion is, according to me, more confusing than helpful and leads to paradigmatical, as well as conceptual, lack of clarity and, thus, vagueness.

## 6. Results

In writing this contribution, I wanted to contribute, in terms of the invitation of the guest editors, with a problematizing approach. The reason for this is the way the concept of 'knowledge' is, unfortunately, often playing a negative and, at the same time, decisive role. It is from a pedagogical, curricular and didactical point of view regularly misused in debates on education in general and, more specifically, in discourses on citizenship education, worldview education and human rights education. And this is to the detriment of a fully articulated, critical, pragmatic pedagogical understanding of different forms of education and their accompanying forms of experience (Miedema and Bertram-Troost 2014; Miedema 2014).

It has become clear that with their approach, Muller and Young are in favor of re- or decontextualized knowledge provided by the academic disciplines. In the practical, academic use of the notion of 'powerful knowledge', it turns out that the concept is used in very context-specific ways in theory formation as well as in research, without having a general or standard conceptual use of the term. In addition, in his own writings, Young has proposed different and differing conceptualizations himself. This, in my view, has led to a conceptual lack of clarity and consistency.

The three presented critics have respectively articulated their criticism from a philosophical (White), a didactical (Carlgren) and a pedagogical perspective (Biesta). White's conclusion with respect to the use of the concept of 'powerful knowledge' by Muller and Young is that in their conceptualization, the relationship between theory and practice is based on precisely a separation of the two. It leads to a sharp distinction between everyday knowledge and disciplinary conceptual knowledge. In his view, we should abandon the use of the term 'powerful knowledge' and use the term 'specialized knowledge'. Carlgren wants to hold practice and theory together, combining a sociocultural and pragmatist approach to knowledge. She proposed the Dewey–Bentley transactional view on knowing and the known, avoiding thus the cartesian, rationalist and dualistic view of Muller and Young. Biesta has criticized Young's social realism with Dewey's transactional realism as a more adequate theory of knowing. Knowledge should indeed be part of the curriculum, however, in Biesta's view on the theoretical basis of Deweyan transactional realism. He has elaborated along these lines on the interrelated notions of 'knowledge' with 'experience' and 'reality' and on the pedagogical and didactical very relevant concept of 'coordination'.

I found the mediating attempts by Reiss (trying to bridge the positions of Young and White) and Nordgren (trying to bridge the positions of Young and Biesta) argumentatively absolutely not convincing. Reiss has tried to find a bridge with Vygotsky's distinction between everyday concepts and scientific concepts. However, unfortunately, Young's Vygotsky interpretation does not hold as could be underpinned by the work of the Vygotsky experts Van der Veer and Valsiner. Nordgren's conclusion that Biesta's reasoning does not jeopardize the idea of powerful knowledge is quite difficult to defend in light of Biesta's position as we have presented in Section 4.

A special case is the chapter by Osbeck, Sporre and Lilja. In fact, the whole discussion on and impact of the notion 'powerful knowledge' came my way when I acted as a critical friend in the project "The child and the curriculum" (Barn och läroplan. Existentiella frågor och skolans svar) of the University of Umeå, Aalborg University and Gothenburg University,

a project led by the researchers Sporre, Kärnebro, Osbeck and Buchardt. During that project, I was, in 2022, invited to comment on what now has become the Osbeck, Sporre and Lilja text from 2023 (Osbeck et al. 2023), with which I dealt in Section 5. While the name of the project is already showing a strong Deweyan pragmatist influence, I especially wondered why the notion of 'powerful knowledge' was still used by the authors, having read the criticism on Young and Muller's and Young's views that the three authors presented themselves, however, without getting rid of the notion of 'powerful knowledge' further on in the presentation of their theoretical and empirical work. In our discussion in 2022, we did not come to a clear conclusion. Afterward, in the correspondence with the authors, it seems that we have approached the debates on the notion of 'powerful knowledge' from different angles—an empirical–curricular one by the authors and the present author's perspective originating and marinated in a critical-hermeneutical-pragmatist-and (neo-)Vygotskian-inspired philosophy of education and philosophy of science.

The core question I formulated in the Introduction was: "How to bring knowledge back in the curriculum and in a way that is philosophically, pedagogically and didactically consistent, compatible and commensurable?". In the end, my plea is that for the sake of conceptual and theoretical clarity, we should abandon the use of the notion 'powerful knowledge' (White) and indeed bring knowledge back into the curriculum, but this via Deweyan pragmatism (Biesta) and social–cultural approaches (Carlgren). Moreover, that we should stick to the use of knowing and the known as a theoretical conceptualization of 'knowledge' because that is compatible with both the pragmatist and sociocultural approaches from a philosophy of science, a philosophy of education and a curriculum theoretical perspective.

## 7. Discussion

Gallie has posed the question of whether or not sharply defined concepts should be characterized as 'essentially contested concepts', and he has described seven characteristics for identifying and understanding whether a concept is essentially contested or not (Gallie 1956). Above I have shown that in my view, 'powerful knowledge' is not such a concept because dealing with the notion 'knowledge' in the curriculum can descriptively be achieved in a clear and distinct, denotative and connotative conceptual way. I do not deny, of course, that there are different connotations of the concept 'knowledge'. Thus, several and also different and differing conceptions of the concept 'knowledge' might be possible (see for the very helpful analytical distinction between concept and conception, Steutel 1991). Reconstructing the different and differing conceptualizations, each having their own described (or rather hidden) theoretical, philosophical, paradigmatical and curricular underpinning, can prevent us from conceptual as well as paradigmatical confusion.

It is still a riddle to me why especially researchers from the Nordic countries working along the lines of pragmatist and sociocultural approaches are so taken to the concept of 'powerful knowledge'. Next to Osbeck, Sporre and Lilja, also Olaf Franck, Gericke, Hudson, Olin-Scheller and Stolare are following this path (Gericke et al. 2018). Maybe this is an interesting issue for further research and reconstruction. Meanwhile, I am glad to see that during the last decades, more and more attention has been paid to didactical and curricular issues inspired by the West European tradition of human science pedagogy in the *Journal of Curriculum Studies* (e.g., Friesen 2020; Klafki 1995; Riquarts and Hopmann 1995), and especially via the excellent reader edited by Westbury, Hopmann and Riquarts *Teaching as a Reflective Practice. The German Didaktik Tradition* (Westbury et al. 2000). Combining the hermeneutical (Geisteswissenschaftliche) tradition, the critical–emancipatory tradition, the pragmatist tradition and the social–cultural tradition in relation to pedagogy and curriculum can at least prevent for dualistic, rationalistic and reductionistic theoretical and practical approaches while giving an adequate place and role for knowledge.

**Funding:** This research received no external funding.

**Institutional Review Board Statement:** Not applicable.

**Informed Consent Statement:** Not applicable.

**Data Availability Statement:** No new data were created.

**Conflicts of Interest:** The author declares no conflict of interest.

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
