# Peer review of "Knowing and the Known: A Philosophical and Pedagogical Critique on the Concept of ‘Powerful Knowledge’"

_socsci, doi:10.3390/socsci12100578_

Round 1

Reviewer 1 Report

Dear Author,

I recommend improving the structure of the post in terms of following the template and supplementing the post with a practical part, as it only acts as a theoretical essay.

Good luck.

Author Response

I have now followed the template as best as I could, due to the fact that this is not an empirical article, but indeed a theoretical essay.  In section 1 now titled "Introduction and method used" I have explained this. In that section I provided an explanation related to the methodological side and have briefly elaborated on the theoretical and practical concept of 'pedagogy'. In the enlarged and differently named section 5 I have addressed the practical, i.e. theoretical and conceptual importance of clear and distinct concept as well as paradigmatical positions, following and paraphrasing Kurt Lewin's maxim "that there is nothing more practical than using clear and distinct (Descartes!) concepts and paradigmatical positions.

Reviewer 2 Report

The main point in this important and well structured article is to question the popularity of the concept of powerful knowledge (PK) launched by Young and Muller given its philosophical and pedagogical presuppositions. This is particularly paradoxical when it comes to research within socio-cultural and pragmatic paradigms which, according to the author, are incompatible and even  incommensurable with the social realism behind PK.

 This is done by first describing three examples of criticisms followed by a questioning of three examples of attempts to overcome the critique and use PK with other theoretical framworks. According to the author these attempts are not convincing and the conclusion is that PK should be abandoned and replaced by a pragmatic conceptualisation in terms of knowing and the known.

It is a well written text that, however, needs to be devloped.  There is a vagueness in the text – above all when it comes to how the attempts to include PK in other frameworks fail. The article would gain a lot from making the arguments more clear and precise. And – if possible – relate the attempts more clearly to the criticisms.

A more developed text would perhaps also solve a problem I have with the text. Namely the lack of distinction between PK as curriculum content and PK as a pedagogical/didactical concept, i.e. how to learn and teach PK. Powerful knowledge as curriculum content (what everyone should be entitled to learn) is not distinguished from PK as a pedagogical/didactic concept. There is a drift in the article towards discussing PK as a pedagogical concept.  Discussing the boundary between everyday and specialized (scientific, powerful) knowledge is different if it concerns content as compared to pedagogy. While White is mainly criticising PK from a curriculum content perspective, Biesta is not.

The important part about ”co-ordination” as a key concept does not refer to PK as curriculum content (what every pupil is entitled to learn) but to a pedagogical situation.

My objection is not that the pedagogical situation is highlighted in the article but that the difference between PK as curriculum content and as a pedagogical concept is not made clear. It weakens the argument.

Author Response

The reviewer has invited me to develop my arguments. I did that in the latest version in which I have brought the different positions far more to the fore, and I also provided in section 1 an underpinning of my (theoretical) method  and gave an elaboration on the concept op 'pedagogy'.

From the comments I notice that the reviewer and I have a different paradigmatical perspectives on the status of pedagogy, didactics and curriculum. That insight - and I really thank the reviewer for addressing this  - is the reason why I have provided in section 1 an elaboration in line with Wolfgang Klafki - whose work has been a source of inspiration for me already from the early 80s on -  in which these 3 concepts are intertwined both in theory and practice. At the end I come back on this with my plea for strenghtening the intertwined approach.

From that perspective coordination is as well a pedagogical, a curricular and a didactical concept. 

Reviewer 3 Report

Dear Author(s)

Hope, my comments listed below can help you to improve the quality of your work.

1. First of all, you need to follow the "INSTRUCTIONS FOR AUTHORS" suggested by the Journal. There, you can find, for example, the following:

§  Research manuscript sections: Introduction, Materials and Methods, Results, Discussion, Conclusions (optional). It means that your manuscript should be structured accordingly.

§  The Reference list should include the full title as recommended by the Chicago style guide.

2. The main research question of your study is not clearly formulated.

3. The main objective(s) of your work should be well presented. It should be clear to readers why your study was conducted and what specific research goal it aimed to achieve.

4. The contribution of your work is not presented clearly. It should represent the "added value" that your research brings to existing knowledge and the broader academic or practical community.

5. The innovation of your work should be presented, explaining the unique and distinctive aspects of the research that differentiate it from previous work and contribute to the advancement of knowledge.

6. You can develop your conclusion by, for example, restating the main objective(s) or research question, providing a concise summary of the most important findings from your research, answering the research question or hypotheses, addressing limitations, and suggesting future work (this could be a sub-section of the conclusion). 

Minor editing of the English language required.

Author Response

ad 1. The reviewer has pointed to the required format of the journal. I had, of course, taken notice of this, however, mine is a pure theoretical/philosophical essay and that differs from articles combining a theoretical and an empirical part. I have experience with both formats.

My reference list indeed includes only full titles.

ad 2. In section 1  I have outlined that the conceptual unclarity in respect to the concept 'powerful knowledge' is the object of my theoretical excercise and that my aim tis o shed some theoretical light on this.

ad 3. In section 1 I have added an explanation related to the methodological approach and elaborated briefly on the theoretical and practical concept of 'pedagogy' in relation with 'curriculum' and 'didactics' in line with Wolfgang Klafki.

ad 4. My contribution is that reconstructing the different and differing conceptualizations (in terms of concept and conceptions) of - in this case - 'powerful knowledge', can prevent us from conceptual unclarity as well as paradigmatical confusion. The added value is sharing my insights with the broader academic and practical communities. It may serve both academic and practical-professional goals.

ad 5. My work is innovative in the sense that it is in case of this theoretical essay  part of the 'handwork' of a theoretician and philosopher to provide conceptual and theoretical clarity. The invitation of the special issue editors was among other aims to contribute with a problematizing approach. I did that along theoretical and philosophical lines.

ad 6. In the enlarged section 5, now titled "Reconstructive results and further conclusions", I present the reconstructive results of my excercise and some further conclusions. I also formulated a new research question, namely why especially researchers in the Nordic countries are so taken to the concept of 'powerful knowledge', while committed to pragmatist and sociocultural approaches. 

Round 2

Reviewer 1 Report

I agree with the modifications.

Author Response

I like to thank reviewer 1 for the helpful criticism I received in the first round, and I'm pleased that my article was accepted in its revised form after the first round of reviews.

Reviewer 3 Report

Dear Author,

1. The developed version of your manuscript shows that some of the suggested comments by the reviewer for improving your work are ignored or not considered. For example, the recommended structure of the Journal is not followed properly.

2. Your manuscript has some development but it is not sufficient. For example, you claim that you used the methods of conceptual analysis, conceptual reconstruction, and systematic literature review. However, no detailed information is presented. For instance, what kind of data were collected and where from, as well as how they were collected and analyzed. Explaining clearly such detailed information will increase the reliability of your work.

Minor editing of the English language is required.

Author Response

ad 1: I have now restructed the article greatly according to the recommended structure of the journal.  Seen the fact that this is a theoretical essay, this was not an easy task, but I think that now the structural train of thought will also be comprehensible for readers who are more familiar with the format of empirical articles.  I have used all my qualitative creativity to make te reviewer satisfied with the result.

ad 2: Via restructuring, there is now a section 2 in which I address the methodological steps I have taken in this qualitative, theoretical essay, and in which I distinguish between data, results, and conclusions. I added also a separate, short discussion section. The result could be found in the changed names of the different sections. Due to the restructuring, the number of sections has also increased.

Based on my own publishing experiences over the last nearly 50 years I can state that for a theoretical essay this is far more than is provided normally. Thus, I really hope that now reviewer 3 can accept the article in its current form.

Thanks for having challenged me to the utmost.

Round 3

Reviewer 3 Report

Dear Author,

Thank you for your contribution and efforts.

 Minor editing of the English language is required.